# Genetic Diversity of Domestic Cat Hepadnavirus in Southern Taiwan

**DOI:** 10.3390/v15102128

**Published:** 2023-10-20

**Authors:** Benji Brayan Ilagan Silva, Jin-Yang Chen, Brian Harvey Avanceña Villanueva, Zi-Ying Lu, Hua-Zhen Hsing, Andrew D. Montecillo, Maya Shofa, Hoang Minh, Jen-Pin Chuang, Huai-Ying Huang, Akatsuki Saito, Kuo-Pin Chuang

**Affiliations:** 1International Degree Program in Animal Vaccine Technology, International College, National Pingtung University of Science and Technology, Pingtung 912, Taiwan; g10985004@mail.npust.edu.tw (B.B.I.S.); bhavillanueva@mymail.mapua.edu.ph (B.H.A.V.);; 2Graduate Institute of Animal Vaccine Technology, College of Veterinary Medicine, National Pingtung University of Science and Technology, Pingtung 912, Taiwan; m11124010@mail.npust.edu.tw; 3School of Chemical, Biological and Materials Engineering and Sciences, Mapúa University, Intramuros, Manila 1002, Philippines; 4Department of Veterinary Medicine, College of Veterinary Medicine, National Pingtung University of Science and Technology, Pingtung 912, Taiwan; 5Microbiology Division, Institute of Biological Sciences, College of Arts and Sciences, University of the Philippines Los Baños, Laguna 4031, Philippines; admontecillo@up.edu.ph; 6Graduate School, University of the Philippines Los Baños, Laguna 4031, Philippines; 7Department of Veterinary Science, University of Miyazaki, Miyazaki 8892192, Japan; 8Graduate School of Medicine and Veterinary Medicine, University of Miyazaki, Miyazaki 8891692, Japan; 9Department of Anatomy and Histology, Faculty of Veterinary Medicine, Vietnam National University of Agriculture, Hanoi 100000, Vietnam; 10Department of Surgery, Chia-Yi Hospital, Ministry of Health and Welfare, Chia-Yi 60069, Taiwan; chuangjp@gmail.com; 11Department of Surgery, Faculty of Medicine, College of Medicine, National Cheng Kung University, Tainan 70101, Taiwan; 12Demin Veterinary Hospital, Kaohsiung 811, Taiwan; 13Department of Pet Care and Grooming, Ta Jen University, Pingtung 912, Taiwan; 14Center for Animal Disease Control, University of Miyazaki, Miyazaki 8892192, Japan; 15School of Dentistry, Kaohsiung Medical University, Kaohsiung 807, Taiwan; 16School of Medicine, Kaohsiung Medical University, Kaohsiung 807, Taiwan; 17Companion Animal Research Center, National Pingtung University of Science and Technology, Pingtung 912, Taiwan

**Keywords:** cat, hepadnavirus, liver, hepatitis, viral, infectious disease

## Abstract

Domestic cat hepadnavirus (DCH) is an infectious disease associated with chronic hepatitis in cats, which suggests a similarity with hepatitis B virus infections in humans. Since its first identification in Australia in 2018, DCH has been reported in several countries with varying prevalence rates, but its presence in Taiwan has yet to be investigated. In this study, we aimed to identify the presence and genetic diversity of DCH infections in Taiwan. Among the 71 samples tested, eight (11.27%) were positive for DCH. Of these positive cases, three cats had elevated levels of alanine transaminase (ALT) and aspartate transaminase (AST), suggesting an association between DCH infection and chronic hepatitis. Four DCH-positive samples were also tested for feline immunodeficiency virus (FIV) and feline leukemia virus (FeLV) coinfection. One sample (25%) was positive for FIV, whereas there was no positive sample for FeLV (0%). In addition, we performed whole genome sequencing on six samples to determine the viral genome sequences. Phylogenetic analyses identified a distinct lineage compared with previously reported sequences. This study highlights the importance of continuous surveillance of DCH and further research to elucidate the pathophysiology and transmission route of DCH.

## 1. Introduction

Domestic cat hepadnavirus (DCH), a novel member of the genus *Orthohepadnavirus* and belonging to the family *Hepadnaviridae*, was recently identified through a virus discovery transcriptomic study on a large cell lymphoma collected from an immunocompromised FIV-positive domestic cat with a hepatic disease [1]. Similar to the other members of the genus *Orthohepadnavirus*, DCH contains partially double-stranded, relaxed circular DNA molecules of about 3.2 kb in length enclosed in an envelope. The genome has four overlapping gene regions encoding the polymerase, surface, core, and X proteins [1].

Previous studies have shown a significant frequency of DCH detection in cats suffering from chronic hepatitis and hepatocellular carcinoma, similar to hepatitis B virus (HBV) infection in humans, which also induces liver damage [2]. Despite these similarities, a review article summarizing the studies currently available suggests that additional information on the pathobiology of DCH infection is necessary to clarify its pathogenicity or direct association with hepatitis in cats [3].

One potential approach for validation involves employing in situ hybridization (ISH) to confirm the presence of DCH in regions of liver inflammation [2]. Moreover, aside from the liver, DCH was also identified in various organs, including the heart, lungs, intestines, kidneys, and spleen, pointing to its broad cellular tropism [1,3,4,5,6]. In addition, repeated testing of cats naturally infected with the DCH showed negative PCR assay results for oral, conjunctival, preputial, and rectal swabs. However, the detection of DCH in the blood and intestinal samples in the same study and other previous studies suggests that the virus may be transmitted via blood or fecal material [3,4,7]. As such, it was previously recommended that DCH be included among agents to be tested in veterinary diagnostic panels, particularly in cases where hepatic disease is clinically suspected and when screening for blood donors [5,8].

Following its initial discovery in Australia, investigations into the prevalence of DCH infections in cats from different regions/countries have been carried out, with positive cases reported in Italy, Thailand, Malaysia, the United Kingdom, Japan, the United States, Hong Kong, and Türkiye [1,2,3,4,5,6,7,8,9,10,11,12,13,14,15,16,17]. The reported detection rates of DCH among these countries showed significant variability, ranging from 0.2% in the USA to about 18.5%, as indicated in one of the reports from Thailand [9,11]. Despite this variability, most studies typically present infection rates of about 10%, such as in Australia (10.4%), Italy (10.8%), Hong Kong (11.1%), Malaysia (12.3%), and Thailand (12.4%) [4,5,6,11]. However, low detection rates were reported in Türkiye (4%), Australia (6.5%), and Japan (> 1.0%) [1,4,5,6,13,15]. Higher detection rates were observed among cats suspected of infectious disease or among cases of chronic hepatitis and hepatocellular carcinoma [2,5]. The detection of antibodies specific to the DCH core protein also resulted in a higher positive detection rate than the detection of viral DNA [16].

In Taiwan, the latest information available through the Pet Registration Information System (https://www.pet.gov.tw/PetsMap/PetsMap.aspx, accessed on 3 October 2023) shows that there are more than 905 thousand registered cats in the system. Moreover, this number excludes unregistered and stray cats. However, despite this significantly large population of domestic cats in Taiwan, it is imperative that there be more information regarding this virus in Taiwan, thus highlighting the importance of DCH investigations.

## 2. Materials and Methods

### 2.1. Ethics Statement

All of the experiments in this study followed the protocols and guidelines approved by the Institutional Animal Care and Use Committee of the National Pingtung University of Science and Technology (NPUST-IACUC), with approval number NPUST-112-079.

### 2.2. Sample Collection

Seventy-one cat blood samples were obtained through multiple clinics in southern Taiwan, following the collection and sample handling guidelines approved by the relevant committee, as aforementioned. Residual, or leftover, samples from five DCH-positive cats were used to test for liver health markers, including alanine transaminase (ALT), aspartate transaminase (AST), and alkaline phosphatase (ALKP). DCH-positive patients were invited for a follow-up health check 2.5 months after the initial detection to monitor changes in their viral loads and liver health markers. Four cats were nominated for the follow-up health check. During these follow-ups, the same DCH-positive cats were also tested for FIV and FeLV coinfection using a SNAP FIV/FeLV Combo Test (IDEXX, Taipei, Taiwan, 99–08354).

### 2.3. DCH Detection by Direct Duplex Quantitative PCR

The detection of the domestic cat hepadnavirus was performed as described previously [18], with minute modifications following the manufacturer’s instructions on the use of Premix Ex Taq™ (Probe qPCR) mix (TaKaRa, Shiga, Japan, RR390L). Briefly, one μL of whole blood sample was added to the PCR reaction mix containing 0.2 μL of each primer (forward and reverse), 0.1 μL of each of the probes at 10 μM concentrations, 10 μL of 2× Probe qPCR mix, 0.4 μL of ROX reference dye, and 7.6 μL of double-distilled sterilized PCR-grade water, making a total volume of 20 μL. The reaction was performed using a StepOne™ real-time thermal cycler system (Applied Biosystems, Singapore). All of the primers, probes, and reaction conditions were as described in [18].

### 2.4. PCR Amplification of the DCH Whole Genome

The total genomic DNA from the serum of the samples that tested positive for DCH was extracted using Viogene Blood and Tissue Genomic Mini (Viogene, Taipei, Taiwan, GG1001), following the manufacturer’s blood protocol. Three sets of primers derived from [12] were used to amplify overlapping fragments covering the whole genome of DCH (Table 1). The PCR reaction, in a final volume of 50 μL, contained 25 μL of P Easy-Pfu 2X PCR SuperMix (AllBio, Taichung, Taiwan, ABTGMBP03-100), one μL of each primer (forward and reverse) at 10 μM concentrations, 10 μL of extracted DNA, and 13 μL of double-distilled sterilized PCR-grade water. The amplification conditions were as follows: an initial denaturation at 94 °C for 5 min, followed by 40 cycles of 94 °C for 30 s, 53 °C for 30 s, and 72 °C for 3 min and 30 s. This was followed by a final extension at 72 °C for 10 min. The amplicons were purified to remove the unused enzymes and dNTPs using the PCR Clean-Up and Gel Extraction Kit (Bio-Helix, New Taipei, Taiwan, PDC01–0100), following the PCR Cleanup protocol as directed by the manufacturer.

### 2.5. Sequencing and Assembly

The purified PCR amplicons were barcoded using the SQK-RBK114.24 kit (Oxford Nanopore Technologies, Oxford, UK, SQK-RBK114.24) and loaded onto a MinION SpotON R10.4.1 FLO-MIN114 flow cell (Oxford Nanopore Technologies, Oxford, UK, R10.4.1 FLO-MIN114). The sequencing was run for 7 h in MinKNOW Software (v.23.04.6) on a MinION MK1b device. Live basecalling was performed using the super accurate 400bps model with default parameters in Guppy v.6.5.7 (Oxford Nanopore Technologies, Oxford, UK).

For each of the barcodes, the basecalled reads located in the fastq_pass folder were combined into one fastq file and mapped against the DCH strain TR-404 genome (GenBank Accession Number: OQ130245.1) using bwa v.0.7.17-r1188. Following this, sequence quality control was performed using Porechop v0.2.4 (https://github.com/rrwick/Porechop, accessed on 3 August 2023) with the default parameters (-discard_middle) [19]. Minimap2 v.2.26-r1175 (-ax map-ont) and Samtools v.1.3.1 (-F) were used to obtain the reads that mapped with the references [20,21]. Cap3 (version date: 2 October 15) was used to perform the assembly, maintaining default settings, and manual curation of the assembled contigs was performed using aligned selected contigs [22]. Alignment was performed in Unipro UGene (v.48.0) using ClustalW, allowing for a default gap opening penalty (15.00) [23]. The assembly was polished in Medaka v1.8.0 (https://github.com/nanoporetech/medaka, accessed on 22 August 2023).

The DCH genomes from the Taiwan strains identified in this study were deposited in GenBank (accession no. OR515499-OR515504). The raw reads were deposited in the NCBI Sequence Read Archive (SRA accession no. SRX21620583- SRX21620588). The Bio-Project and BioSample accession numbers are PRJNA1011820 and SAMN37224405 to SAMN37224410, respectively.

### 2.6. Phylogenetic and Recombination Analyses

Alignments of the recovered genome and protein sequences against available DCH and other hepadnavirus sequences were performed using Clustal (v.2.1). Maximum likelihood trees were constructed in IQTREE v.2.2.2.3 using the default settings, allowing the ModelFinder to select the suitable substitution model for the construction of the phylogenetic trees [24,25].

Using all available DCH sequences, the presence of putative recombination sites within the genomes of the Taiwan DCH strains was examined using different models (RDP, GENECONV, Bootscan, MaxChi, Chimaera, SiScan, 3Seq, LARD) implemented in RDP5 (version 5.23) [26]. The recombination events with *p*-values ≤ 0.05 for at least three different models were considered valid.

## 3. Results

Seventy-one blood samples were analyzed, of which 13 were from Tainan City, 45 were from Kaohsiung City, and 13 were from Pingtung County (Figure 1). Thirty-seven (37, 52.11%) samples came from female cats, equalizing the sexes within the sample population. The majority (58 out of 71, 81.69%) of the sampled cats were neutered, whereas approximately a quarter were exclusively residing indoors (15/71, 21.13%). The age range of the tested cats spanned from 6 months to 9 years, with a median age of 3 years. Most of these cats (65 out of 71, 91.55%) appeared to be in good health, showing no clinical signs of any disease, whereas the remaining six displayed non-specific symptoms, mainly due to poor appetite.

In the 71 blood samples detected by quantitative polymerase chain reaction, eight (11.27%) were positive for DCH. Four out of the eight cases detected were from Kaohsiung City, two cases were from Tainan City, and the other two cases were from Pingtung County (Figure 1). The quantitative estimates of the viral load in these samples varied, ranging from 3 × 10^5^ to 6 × 10^8^ copies/mL (Table 2).

Furthermore, five of the eight blood samples were tested for a selection of blood biochemistry markers related to liver function. Of these, three (3/5; 60%) of the tested samples displayed high levels of ALT and AST; however, all samples produced normal results for ALKP (Table 2). There were no apparent symptoms observed in any of the cats that tested positive for DCH. Moreover, no DCH infection was detected among the cats displaying poor appetite.

In four of the eight cases, we were able to perform follow-up health checks 2.5 months after the DCH detection. The viral load and liver health markers were remeasured. The results are presented in Figure 2. They showed that all tested cats displayed an increased viral load relative to the initial measurement at day 0. With the exception of case 23-05-30-004, three of the four cats displayed increased AST levels; two cases (cases 23-05-30-005 and 23-05-30-019), which were previously within the normal AST range, were found to have abnormal AST levels during the follow-up. In addition, ALT levels for two cases (cases 23-05-30-004 and 23-05-30-017) decreased, with the latter returning to within the normal range. The other two cases, however, were observed with increased ALT levels. with case 23-05-30-005, which was previously within the normal ALT range, was observed to be outside the normal ALT range during the follow-up. Moreover, all cats continued to produce normal test results for ALKP 2.5 months after their positive DCH detection. Additionally, the DCH-positive cats were tested for feline immunodeficiency virus (FIV) and feline leukemia virus (FeLV) during the follow-up. Among these, only one case tested FIV-positive, which is case number 23-05-30-017 (1/4; 25%), and none were detected with FeLV (0/4; 0%) (Table 2).

For further analysis, the total DNA was extracted from six selected sera from the positive blood samples. The whole genome of DCH was amplified using three primer pairs. Purified PCR amplicons were quantified by nanodrop, and the three amplicons from a single sample were combined in equal amounts and processed for Oxford Nanopore Technologies sequencing, following the rapid barcoding kit protocol (SQK-RBK114.24). A total of more than 150,000 reads (N50 = 933 b) were generated, of which more than 100 Mb were retained following the adapter trimming.

Complete closed genomes were successfully amplified and sequenced from six blood samples. The genomes were 3184 bp in length. Homology analysis using BLASTn search against the NCBI nr database revealed that, at the full genome level, five Taiwan strains (DCH/NPUST-001/TWN/2023 to DCH/NPUST-005/TWN/2023) displayed the greatest similarity, approximately 98.7%, to DCH strain TR-404 from Türkiye, whereas another strain (DCH/NPUST-006/TWN/2023) had the highest similarity to PK98-B/THA/2022 from Thailand with a percent ID of 99.1%. In addition, the Taiwan strains shared 97.5% to 100% nucleotide pairwise identities with each other (Appendix A). Collectively, these observations suggest the presence of multiple and diverse strains of the DCH in Taiwan. In this study, no evidence of genetic recombination events was detected in the genomes of the Taiwan strains when queried together with all other available DCH genome sequences. Examination of the preS1 amino acid sequences confirmed the previously observed conservation of this region among DCH strains, including the Taiwan strains (Appendix A) [27].

Our phylogenetic analysis, using the maximum likelihood method on all of the available complete sequences of DCH and the genomes of other members of the *Orthohepadnavirus*, *Metahepadnavirus*, *Herpetohepadnavirus*, and *Avihepadvirus* genera, supports observations reporting the clustering of the available sequences broadly according to their genera (data/ML tree not shown), as previously reported [6]. In addition, among the sequences of DCH, the DCH/NPUST-006/TWN/2023 strain, grouped with the Hong Kong (HK03/2020/16) and Thai (KB18-B/THA/2022 and PK98-B/THA/2022) strains, forms a distinct phylogenetic lineage that is sister to the clade to which the prototypic Sydney2016 strain belongs (Figure 3). The other Taiwan strains, however, formed a distinct clade occupied only by the sequences from this study and sister to the clade to which most of the genomes recovered from Türkiye belong.

Generally, DCH/NPUST-006/TWN/2023 belongs to clade A1, as proposed previously by other authors [13], whereas the rest of the Taiwan strains are grouped closer to the proposed clade A2, under genotype A. However, with the addition of more sequences, particularly those from the Türkiye and Taiwan strains, our phylogenetic analysis of the currently available DCH sequences, implemented by constructing a maximum likelihood tree using TIM+F+I+R2 as the best fit and chosen model according to the Bayesian information criterion, did not reveal a robust branching support for a clear distinction between the proposed clades A1 and A2 (Figure 3, Appendix A). As such, the assignment of clades within genotype A may need revisiting. Notably, the Taiwan strains DCH/NPUST-001/Taiwan/2023 to DCH/NPUST-005/Taiwan/2023 may represent a novel clade, given the topology of the tree from the current analysis. However, more information would be needed to confirm this. Rara strain from Japan, on the other hand, remains as the only member of DCH genotype B.

The pairwise identity scores using all available sequences of the four DCH proteins revealed the highest conservation in the core protein sequences, with percent IDs ranging from 97.7 to 100. Moreover, X protein showed the greatest variation, with percent IDs ranging from 75.2 to 100. The DCH strains’ polymerase and surface protein percent IDs ranged from 85.4 to 100. The phylogenetic relationships among the Taiwan strains, particularly the divergence of DCH/NPUST-006 from the other strains, as observed in Figure 3, are strongly supported by the phylogenetic trees based on sequences of the core, polymerase, surface, and X protein sequences (Appendix A).

The maximum likelihood trees of both the polymerase and surface proteins (Appendix A) generally show the two subdivisions of genotype A as proposed previously by other authors [13]. However, as with the whole genome sequence tree, there is no robust bootstrap support for this proposed subdivision. Furthermore, such a distinct division is not apparent in the phylogenetic trees based on the core and X proteins (Appendix A). Notably though, based on the phylogenetic tree constructed based on the core protein sequences, strains DCH/NPUST-001/TWN/2023 to DCH/NPUST-005/TWN/2023 were separately grouped with a strain each from Italy and Thailand in a basal clade separate from all other genotype A strains (Appendix A).

## 4. Discussion

In recent years, an increasing number of countries have reported the detection of DCH, a novel infectious viral agent affecting domestic cats. To elucidate the presence and genetic diversity of DCH in Taiwan, we screened blood samples from 71 cats and demonstrated that eight (11.27%) cats were DCH-positive. Despite a limited number of samples, this study revealed the presence of DCH in Taiwan for the first time. Additionally, because we were only able to analyze samples collected in the southern areas of Taiwan, more surveillance studies are needed to reveal the accurate DCH infection prevalence in Taiwan. Furthermore, our phylogenetic analyses demonstrated that the DCH strains in Taiwan may represent a distinct lineage compared with previously reported sequences. These findings highlight the importance of continuous surveillance of DCH and further research to elucidate the pathophysiology and transmission route of DCH.

Consistent with previous studies, our observations suggest no significant association between sex and DCH detection [5,6]. In addition, it is interesting to note that while an earlier study observed a higher DCH detection rate among older cats (ages 5–9 years), only one (1/6, 16.7%) of the detected cases in this study was from the same age group, but the majority of the DCH-positive cases were detected among adult cats (>two years), which is consistent with previous reports [6,9]. Furthermore, all positive cases were detected in cats that were allowed access outside of the house or were not exclusively residing indoors, thus opening a possibility for transmission from other cats. It has been previously hypothesized that the horizontal transmission of DCH may happen through exposure to contaminated blood or fecal material [3,4,7]. Regarding health status, DCH infection was not detected among cats presenting symptoms of an ill condition.

At Day 0, all DCH-positive cats in this study were healthy and displayed no disease or infection symptoms. However, in previous reports, the levels of ALT and AST were elevated in about half (3/5, 60%) of the samples tested for markers of liver condition [5,6]. The virus titers of the samples with elevated hepatic markers are >10^4^ genomic copies/mL, indicative of an acute infection or the active chronic stages of the disease if the HBV titer threshold in humans is applied in DCH infections [5]. However, the samples with elevated markers in this study do not correspond to higher virus titers. Moreover, and potentially complicating this further, all of the samples displayed virus titers > 10^4^ genomic copies/mL threshold, but two (2/5, 40%) displayed normal hepatic markers. As a whole, the current study echoes the recommendation that DCH be included in veterinary diagnostic panels, particularly in cases of clinically suspected hepatic diseases and in the screening of blood donors [5,8].

Likewise, during the follow-up health check 2.5 months after detection, the changes in AST and ALT have no correlation with the changes in viral load. Although the available information showed that AST tends to increase with viral load, case 23–05–30–004 proved to be an exception that was observed to have the most significant decrease in AST levels, even approaching the normal range whilst also displaying a significantly increased viral load.

A positive correlation of DCH with FIV and FeLV was previously reported in the literature [3,4,7]. In this study, among DCH-positive cats, one tested positive for FIV (1/4, 25%), whereas none tested positive for FeLV (0/4, 0%). Considering the few DCH-positive sampling sizes, further extensive detection across Taiwan is needed to investigate the coinfection of FIV and FeLV with DCH. However, collecting blood samples from cats for detection and continuous monitoring of the virus is challenging. Thus, wide-ranging detection of DCH and its coinfection with FIV and FeLV is necessary to evaluate its emerging threat to cats in Taiwan.

Furthermore, this study confirmed the conservation of the DCH preS1 amino acid sequence among DCH strains, including in Taiwan strains, as previously observed [27]. This observation is particularly relevant in relation to a recent study that demonstrated how the HBV- and DCH-derived preS1 peptide efficiently bound to the sodium/bile cotransporter (NTCP) in several mammalian species, including humans. Additionally, it was previously identified that position 158 of NTCP proteins determines species-specific binding of the preS1 peptide and that Myrcludex B, a known HBV entry inhibitor, was also shown to block the binding of the DCH preS1 peptide to NTCP. Overall, these findings suggest a shared nature of the binding mechanism of DCH- and HBV-derived preS1 peptides to the NTCP receptor and raise the potential of DCH for interspecies and even zoonotic transmission [27]. Therefore, more studies must be conducted to establish the pathobiology of the virus and the pathophysiology of the infection [3,27]. Isolation and propagation of infectious DCH particles would also permit further studies to directly examine the species tropism of DCH in various hosts or cells, or under positive or negative host factors that may influence viral replication [27].

Lastly, the genetic diversity observed in this study also suggests the need to continuously surveil this virus among Taiwan’s cat populations and monitor for the introduction of new strains or the potential emergence of recombinants, as observed in another report [4].

## Figures and Tables

**Figure 1 viruses-15-02128-f001:**
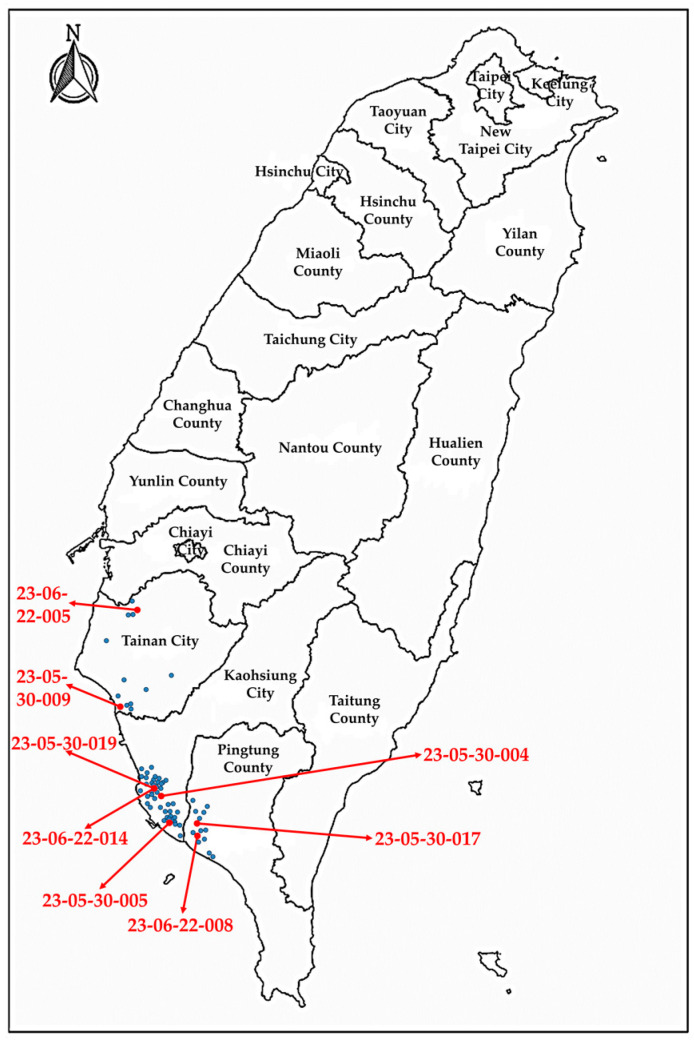
The geographical location of the DCH-positive samples (red arrows) and the DCH-negative samples (blue dots). The map of Taiwan was sourced from the National Land Surveying and Mapping Center (NLSC).

**Figure 2 viruses-15-02128-f002:**
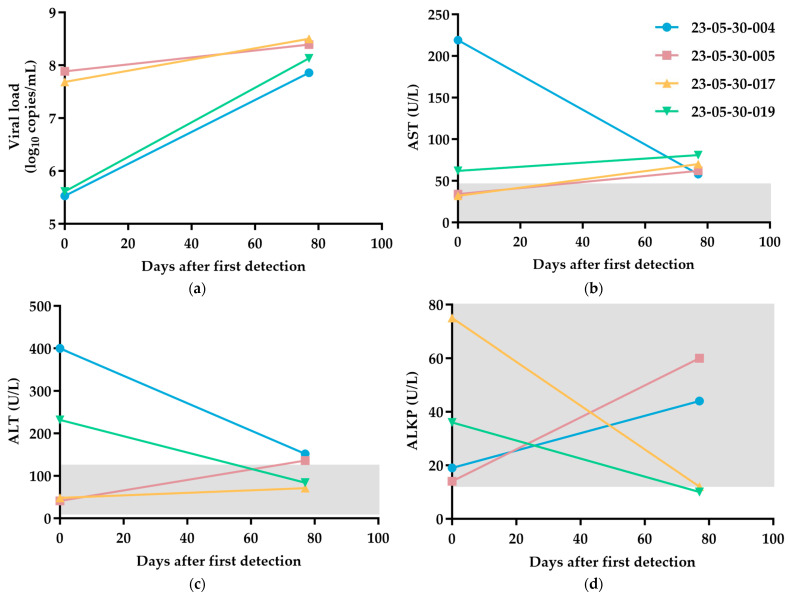
The changes in (**a**) viral load, (**b**) AST, (**c**) ALT, and (**d**) ALKP levels in DCH-positive cats 2.5 months after the first detection. In (**b**–**d**), the values within the normal range are indicated by gray shading. Normal Range: AST 0–48 U/L; ALT 12–130 U/L; ALKP 14–111 U/L.

**Figure 3 viruses-15-02128-f003:**
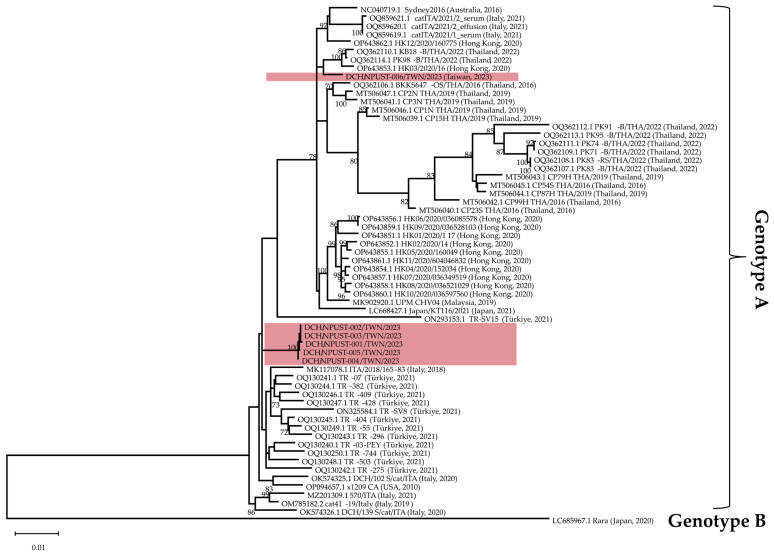
The phylogenetic tree was constructed using complete genomes of DCH collected from Taiwan and available sequences from GenBank. The tree was created by constructing a maximum likelihood tree with TIM + F + I + R2 as the best fit and chosen model according to the Bayesian information criterion using IQTREE v.2.2.2.3. Bootstrap values above 70% are shown. The scale bar indicates the nucleotide substitution per site. Taiwan strains are indicated by red shading.

**Table 1 viruses-15-02128-t001:** Primers used for the amplification of the DCH whole genome.

Primer Set	Forward Primer	Reverse Primer	Product Size (bp)
F1R2	5′-ACTCTCAAACAGGGAACATTCGT-3′	5′-ATTCCAATAGCAGATCACGTAG-3′	1243
F2R3	5′-AATTCTCCAAAGGCTAACAGGTTTA-3′	5′-CAAGACAGTATGTTGTCCAAAAGTG-3′	1267
F4R6	5′-GAAGAGGAACTTACAGGTAGGGAAC-3′	5′-CATCCATATAAGCAAACACCATACA-3′	1913

**Table 2 viruses-15-02128-t002:** Collated information for the sampled domestic cats that tested positive for domestic cat hepadnavirus.

Case Number	Age (Years)	Sex	Viral Load * (Copies/mL)	ALT ^1,^* (U/L)	AST ^2,^* (U/L)	ALKP ^3,^* (U/L)	FIV/FeLV	Strain Code (GenBank Acc. No.)
23-05-30-004	2	M	~3 × 10^5^	**400**	**219**	19	Negative/Negative	DCH/NPUST-001/TWN/2023 (OR515499)
23-05-30-005	1	M (neutered)	~7 × 10^7^	41	34	14	Negative/Negative	DCH/NPUST-002/TWN/2023 (OR515500)
23-05-30-009	1	F (neutered)	~7 × 10^7^	**170**	**78**	19	-	DCH/NPUST-003/TWN/2023 (OR515501)
23-05-30-017	2	M	~4 × 10^7^	48	32	75	Positive/Negative	-
23-05-30-019	4	F (neutered)	~4 × 10^5^	**232**	**62**	36	Negative/Negative	DCH/NPUST-004/TWN/2023 (OR515502)
23-06-22-005	4	F (neutered)	~3 × 10^8^	-	-	-	-	DCH/NPUST-005/TWN/2023 (OR515503)
23-06-22-008	5	F (neutered)	~3 × 10^6^	-	-	-	-	-
23-06-22-014	2	F	~6 × 10^8^	-	-	-	-	DCH/NPUST-006/TWN/2023 (OR515504)

* Measured at first sampling. Normal range: ^1^ 12–130 U/L; ^2^ 0–48 U/L; and ^3^ 14–111 U/L. The values outside the normal range are indicated in bold. (-) The information is not available, or the test was not performed.

## Data Availability

All data generated and analyzed in this study are included in this article.

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
