# Peer review of "Genetic Diversity of Domestic Cat Hepadnavirus in Southern Taiwan"

_viruses, 2023, doi:10.3390/v15102128_

Round 1

Reviewer 1 Report

The authors have elucidated, for the first time, the distribution status of DCH in domestic cats in southern Taiwan, exploring its correlation with clinical symptoms and, further, its association with FIV and FeLV. Additionally, through phylogenetic analysis of DCH, the authors have scrutinized how closely related the DCH in this study is to other previously reported hepadnaviruses. It is my hope that the following comments will assist in refining the manuscript.

Comment 1.  It is probable that direct contig output from BWA mapping results (in Bam format) is unachievable. Which software was employed for outputs from BWA? (bcftools and samtools?)

Comment 2.  Please specify the anticipated amplification product size of the primer set used during whole-genome amplification. Also, I presume that the DNA concentration extracted from the gel was extremely low, was it not? This could be due to dsDNA being sheared as it passes through the gel. If so, I would recommend improving the method in future applications.

Comment 3.  Regarding Comment 2, outputting 150,000 reads using Nanopore MinION seems quite scant. Generally, a MinION flow cell (presuming champion data) can yield outputs of 20Gb, approximating around 2.4 million reads of data. While 150,000 reads may suffice for this analysis, it seems imperative to assess the run. Tools such as Nanostats, Nanoplot, or nanoQC could be utilized for this purpose.

Comment 4.  If the zoonotic potential of DCH is to be discussed, wouldn't a preS1 sequence analysis of the DCH analysed here be necessary?

Comment 5.  In the methods section, it appears that 1 µl of whole blood is added directly to the PCR, is this correct? In general, I feel that blood contains PCR inhibitors, which reduces detection sensitivity.

Comment 6.  Regarding the two samples that could not be wholly amplified despite being positive in qPCR, does this mean that no genes were amplified in any PCR with the primer sets? If even a portion is amplified, it can firmly assert the presence of the virus in the sample.

Comment 7.  In the results section, where you explain using Fig 3, you describe Clade A1 and A2, but how about reflecting this clade division in each respective Fig?

Author Response

Dear Reviewer,

Please see the attachment file for the Author's responses. 

Thank you very much for your time.

Reviewer 2 Report

The submitted manuscript “Genetic diversity of domestic cat hepadnavirus in Taiwan” by Silva and collaborators reports for the first time the presence of domestic cat hepadnavirus in Taiwan along with a phylogenic analysis of the strains identified. Overall, the manuscript is well written and divided in 4 sections, including an short introduction, a clear material and methods, a concise results section, and a short discussion. Based on the reviewer opinion, there are concerns that the authors need to address before considering publication:

1.     The reviewer's main concern is the number of samples collected and the geographical location from which they were collected. Firstly, analysis of blood samples from just 71 cats is not sufficient to assess the prevalence of a virus in Taiwan. Secondly, we don't know exactly where all the samples were collected (positive and negative). The location of negative samples should be indicated on figure 1, using blue dots for example. In this manuscript, all samples were collected in southern Taiwan (around Kaohsiung). However, Taiwan's population is mainly divided into three regions (Kaohsiung (south), Taichung (center) and Taipei (north)). To assess the prevalence of CDH in Taiwan, samples should be taken "equivalently" in these three regions and not just in the south. The authors should change the name of the title to "southern Taiwan", or they should collect more samples in the other regions of Taiwan in order to use the word "prevalence", as the number of samples collected is low.

2.     The introduction is extremely brief. The authors remain evasive on the importance of this virus, the diseases it causes and have not indicated the actual classification of DCH (Clades A1, A2 and B), which makes the results section confusing. In addition, although little is known about DCH, the link and similarity with HBV should be added.  Line 57: HBV should be Hepatitis B Virus (HBV). The authors have listed countries where HDC prevalence has been performed, but it would be useful to include prevalence data. Line 88, all quantitative PCR are real-time so the “real-time” need to be removed.

3.     The authors indicated that one ul of whole blood was used as template DNA for each sample, corresponding to approximately 1000 PBMCs. The reviewer believes that this is a very small amount of blood and that some positive samples may therefore go undetected.

4.     How the samples were selected for this study. Did the authors take all the blood samples collected without any exclusion/inclusion criteria. Are the 6 cats presenting poor appetite are the cats positives for DCH?

5.     Table 2: The value outside the normal range are not indicated in bold.

6.     Why the sex of the case number 23-06-22-114 is not indicated?

7.     Line 170: Is “Five” should be a “Eight”?

8.     Why did the authors sequence the entire HDC genome for only six samples, and not for the eight positives?

9.     The clades A1 and A2 should be included in the Figure 3 and 4.

Author Response

(The authors gave the same response as above.)

Reviewer 3 Report

This is an interesting descriptive study with limited number of subjects involved. It is known that the island of Taiwan is populated with cats and the study could be improved providing information on the population density of owned cats. This information may help readers understanding samples representativeness of the study. In the "Sample Collection" paragraph it would be advisable to be precise and indicate the number of samples. I would recommend removing the concept of zoonosis from the abstract as this hypothesis could trigger alarmism and needs to be furtherly confirmed. Moreover, I would leave the concept of zoonosis in the "discussion" where the topic should be  explained more in-depth.

Author Response

(The authors gave the same response as above.)

Reviewer 4 Report

The article "Genetic Diversity of Domestic Cat Hepadnavirus in Taiwan" by Benji Brian Ilagan Silva and colleagues examines the prevalence of DCH in the domestic cat population of Taiwan. This is the first study of its kind, so it is of particular interest. The manuscript describes the results of sample analysis, including bioinformatics analysis. The latter clarified evolutionary relations between known DCH groups and Taiwanese viruses. The correlations obtained are of particular interest, but the sample used is not very large. The manuscript is well structured. However, some improvements can be made to improve the article.

The introduction could have better described the results of studies conducted in other regions.

The Materials and Methods section could better describe the bioinformatic methods used, including:

Line 122 - missing reference (https://doi.org/10.1093/bioadv/vbac085?) for Porechop.

Line 123 - missing settings and reference for Cap3.

Line 125 - incomplete settings, missing reference for ClustalW, missing reference for Ugene.

Line 136 - missing settings and reference for IQTREE.

The Results section provides an accurate analysis, but it can also be improved.

Line 191-199 - I would suggest building a cluster heatmap that can clearly show the level of pairwise identity between sequences.

Figure 4 – Perhaps these figures, which support the findings, could be moved to the Supplementary Materials.

Author Response

(The authors gave the same response as above.)

Round 2

Reviewer 2 Report

Authors have adequately responded my questions. There are no further questions raised by me. 

Reviewer 4 Report

All comments were answered correctly, and the manuscript appears to be suitable for publication in the journal Viruses.